# The Rise of Hypothesis-Driven Artificial Intelligence in Oncology

**DOI:** 10.3390/cancers16040822

**Published:** 2024-02-18

**Authors:** Zilin Xianyu, Cristina Correia, Choong Yong Ung, Shizhen Zhu, Daniel D. Billadeau, Hu Li

**Affiliations:** 1Department of Molecular Pharmacology and Experimental Therapeutics, Mayo Clinic College of Medicine and Science, Rochester, MN 55905, USA; xianyu.zilin@mayo.edu (Z.X.); correia.cristina@mayo.edu (C.C.); ung.choongyong@mayo.edu (C.Y.U.); zhu.shizhen@mayo.edu (S.Z.); billadeau.daniel@mayo.edu (D.D.B.); 2Department of Biochemistry and Molecular Biology, Mayo Clinic College of Medicine and Science, Rochester, MN 55905, USA; 3Department of Immunology, Mayo Clinic College of Medicine and Science, Rochester, MN 55905, USA

**Keywords:** Artificial Intelligence, machine learning, domain knowledge, system biology, oncology

## Abstract

**Simple Summary:**

This review introduces a new class of Artificial Intelligence (AI) algorithms called hypothesis-driven AI. We elaborate on how this new class of AI methods differs from conventional AI using published AI algorithms and illustrate the power of hypothesis-driven AI in making new discoveries in cancer research. Finally, we outline the ingredients needed to overcome limitations and expand hypothesis-driven AI in the near future.

**Abstract:**

Cancer is a complex disease involving the deregulation of intricate cellular systems beyond genetic aberrations and, as such, requires sophisticated computational approaches and high-dimensional data for optimal interpretation. While conventional artificial intelligence (AI) models excel in many prediction tasks, they often lack interpretability and are blind to the scientific hypotheses generated by researchers to enable cancer discoveries. Here we propose that hypothesis-driven AI, a new emerging class of AI algorithm, is an innovative approach to uncovering the complex etiology of cancer from big omics data. This review exemplifies how hypothesis-driven AI is different from conventional AI by citing its application in various areas of oncology including tumor classification, patient stratification, cancer gene discovery, drug response prediction, and tumor spatial organization. Our aim is to stress the feasibility of incorporating domain knowledge and scientific hypotheses to craft the design of new AI algorithms. We showcase the power of hypothesis-driven AI in making novel cancer discoveries that can be overlooked by conventional AI methods. Since hypothesis-driven AI is still in its infancy, open questions such as how to better incorporate new knowledge and biological perspectives to ameliorate bias and improve interpretability in the design of AI algorithms still need to be addressed. In conclusion, hypothesis-driven AI holds great promise in the discovery of new mechanistic and functional insights that explain the complexity of cancer etiology and potentially chart a new roadmap to improve treatment regimens for individual patients.

## 1. Introduction

Cancer is a complex disease with a wide array of factors contributing to its etiology. Understanding its underlying mechanisms requires sophisticated computational approaches to uncover not only the genetic permutations contributing to uncontrolled cell growth, but also the cellular and systemic factors that contribute to cancer development and response to therapy. Years of research have shown that cancer is not just a disease of genes but rather a disease of systems [1], including epigenetic modifications, alterations in signaling pathways, tumor microenvironment interactions, immune system responses, and lifestyle factors. This indicates that the etiology of cancer cannot be merely attributed to aberrations of a number of genes, but rather involves a broad array of biological factors, including the microenvironment [2] and microbiome [3].

Oncology research confronts a number of significant hurdles, including reliable tumor detection, patient stratification, risk assessment and therapy matching. Although enhancing early cancer detection methods is crucial, genetic and molecular complexities surrounding tumor growth dynamics and the onset of metastasis still pose formidable challenges. For example, the intricate interplay between tumor development and metastasis requires further investigation to refine early detection strategies [4].

Another pivotal challenge in oncology lies in tumor targeting. In oncogene-driven cancers, robustly identifying and effectively targeting actionable genomic alterations is important. However, druggable genomic alterations exist in only a small fraction of patients with specific tumor types, which restricts the performance of clinical testing in biomarker-driven trials. In addition, the clinical interpretation of large genomic datasets remains a formidable barrier to their widespread clinical application. Moreover, tumor heterogeneity and the emergence of acquired drug resistance are significant barriers that undermine the success of precision oncology approaches [5,6]. Regarding immuno-oncology, challenges persist in the successful targeting of solid tumors and the achievement of enduring survival benefits with immune checkpoint inhibitors, monoclonal antibodies, chimeric antigen receptor (CAR) T-cell therapy, and beyond. Currently, only a minority of patients experience long-term benefits from these therapies, and the absence of robustly validated predictive markers complicates patient selection and treatment optimization [7]. Overcoming these hurdles may require the development of drug combinations targeting multiple hallmarks of cancer, which indicates a need for a multifaceted approach.

The burgeoning of next-generation sequencing technologies in the past decade, especially single-cell sequencing, has facilitated the generation of large amounts of cancer omics data at an unprecedented scale, together with high-resolution imaging data. Such high-dimensional omics data are inherently complex, and biologically meaningful signals are not necessarily detected using linear models and conventional statistical analyses. Artificial intelligence (AI) has the power to uncover the nonlinear associations of meaningful signals and, as such, has gained momentum in oncology research in recent years [8].

The rapid evolution of AI technologies, especially deep learning, has emerged as a transformative force to revolutionize the way we study cancer, including target identification and drug response prediction. Although these AI methods are powerful enough to classify cancer types, stratify patients, and predict outcomes, any meaningful biological signals learned by AI models are usually obscure to the users. As such, feature selection methods, such as recursive feature elimination and information gain, have been devised to identify the data attributes, such as genetic mutations, that contribute to the performance of AI models. However, feature selection remains external to the learning processes of AI models and what useful “knowledge” AI models have learned is still inaccessible. Methods such as decision trees and explainable AI (XAI) [9] have the potential to indicate how AI models learn and make decisions. Nonetheless, we still lack the information needed to discern how the meaningful data attributes relate and associate to one another, which is critical for formulating testable hypotheses to explain the pathogenesis of cancers.

In this review, we coined the term “hypothesis-driven AI” to refer to a new emerging class of AI for which the learning algorithms are designed based on novel scientific hypotheses. Although some of these AI algorithms have not been well tested, they nonetheless provide a new avenue for the use of AI technologies in oncology research. We will first discuss how hypothesis-driven AI differs from conventional AI and then provide examples of existing AI tools, whose learning algorithm design aligns to a certain extent with the proposed configuration of hypothesis-driven AI. We will also discuss their utilities and potential challenges. Finally, we offer our view on the future of hypothesis-driven AI in oncology research and discuss how this new AI class can revolutionize individualized and precision medicine.

### 1.1. What Is Hypothesis-Driven AI

The differences between hypothesis-driven AI and conventional AI are summarized in Figure 1. In brief, conventional AI, be it supervised or unsupervised, utilizes existing generic learning algorithms that aim to optimize the mathematical parameters that best fit input data without the incorporation of prior knowledge in the training processes (Figure 1A). For supervised learning, data are labeled with known classes for classification models or quantitative measures, such as responsive drug dosages, for regression models. For unsupervised learning, such as hierarchical clustering, k-nearest neighbor, and self-organizing maps, the similarity of samples is the basic assumption where distance metrics such as the Euclidean distance are employed. Feature selection methods are often used to identify the data attributes (also called features) that contribute to the performance of AI models. As shown in Figure 1, in the algorithm design and training process of conventional AI, the inherent associations of selected features that explain the properties of data are often obscured.

Contrary to conventional AI, the design of hypothesis-driven AI algorithms requires the incorporation of domain knowledge and hypotheses (Figure 1B). This indicates that the learning algorithms of hypothesis-driven AI are more flexible and open to manipulation based on specific hypothesis settings. Unlike conventional AI, the design of hypothesis-driven AI models demands the ingenuity of designers while offering flexible frameworks. In principle, hypothesis-driven AI can be designed based on any existing AI major domains, including artificial neural networks (ANNs), support vector machines (SVMs), random forest, and genetic algorithms. Both supervised and unsupervised learning processes are applicable. Feature selection may or may not be included. However, the resulting trained hypothesis-driven AI will indicate the inherent structures of the meaningful data attributes that explain the behavior of systems, enabling researchers to formulate testable mechanistic models. Table 1 further summarizes the differences between conventional and hypothesis-driven AI.

### 1.2. Why Do We Need Hypothesis-Driven AI in Oncology Research

In order to overcome the complexity of cancer data, researchers have developed a variety of AI tools to help with cancer diagnosis and treatment. As mentioned, a large number of AI algorithms have been designed and applied to analyze the patterns and relationships in large datasets for image recognition, genomic analysis, and clinical data interpretation. However, most conventional AI models are considered as a “black box”, meaning that their decision-making process is not interpretable, and the rationale behind their results is not understandable to researchers or clinicians. Apart from that, these models are highly dependent on the quality and quantity of the dataset they use. If the dataset is not large or diverse enough, noise or bias can be easily introduced, and overfitting issues become very common. Moreover, since these models often lack focus on a specific scientific or clinical question, the results generated may not offer deeper insights to formulate testable mechanistic hypotheses. Lastly, running non-hypothesis-driven data analyses via routine training procedures is usually less efficient, as the researchers are blinded to the way in which to take the most from the wealth of information embedded in the data. For instance, conventional analytical pipelines usually look for differentially expressed genes or mutated genes in cancer data. Such analytical procedures are mainly based on the assumption that differentially expressed genes and mutated genes are key players in cancer development. However, as we will illustrate, these assumptions are not always true. Hence, there is an unmet need to incorporate new perspectives and hypotheses into the design of AI algorithms in order to better steer AI-based cancer research, as summarized in Figure 2.

One salient advantage of hypothesis-driven AI in oncology is that it offers a targeted and informed approach to addressing many of the challenges mentioned above. Compared to conventional AI, hypothesis-driven AI is able to perform focused investigations by centering on specific hypotheses or research questions and thus enables the use of prior knowledge to guide its exploration. This approach can generate more interpretable and explainable results compared to conventional AI tools, since the underlying hypotheses provide a mechanistic framework for understanding the logic behind certain predictions or outcomes. Apart from that, hypothesis-driven AI tends to use data resources more efficiently. Since it allows researchers to concentrate their computation on areas of particular interest, hypothesis-driven AI can reduce the need for extensive data and computational resources. It also encourages the integration of domain-specific knowledge to generate meaningful insights within a specific clinical context. Moreover, this approach allows researchers to test hypotheses and validate them via AI-based thought experiments, which in turn guides future experimental designs.

By integrating computational methodologies with compelling hypotheses, these AI tools can significantly impact patient outcomes in oncology. For instance, AI algorithms guided by hypotheses can transform complex data into patterns indicative of early-stage cancers, potentially revolutionizing early cancer detection methods. Furthermore, for tumor targeting, hypothesis-driven AI can play a pivotal role in prioritizing actionable genomic alterations for further investigation. By integrating domain knowledge and informed hypotheses, AI models can sift through genomic datasets to identify alterations with the highest potential for clinical impact, which could also mitigate the limitations caused by the complexity of druggable alterations and the challenges in interpreting clinical outcomes. Additionally, hypothesis-driven AI can aid in understanding the complex dynamics of tumor heterogeneity and acquired resistance. By modeling and simulating the evolutionary processes within tumors, these new AI models can help validate hypotheses about the key drivers of resistance and potential therapeutic strategies able to counteract them. Below, we provide examples of AI methodologies for which the design of their learning algorithms to certain extent is aligned with the configuration of hypothesis-driven AI. We also show case applications in different aspects of oncology while summarizing the underlying hypothesis of each of these tools in Figure 3. These AI tools are the predecessors of hypothesis-driven AI and insightful examples to illustrate the future of hypothesis-driven AI methods are given.

## 2. Recent Developments and Application of Hypothesis-Driven AI in Oncology

### 2.1. Tumor Classification

Classifying tumors based on their molecular and pathological features is important when aiming to inform clinicians about the type and stage of a tumor and plans for treatment. One key challenge is to classify cancers of unknown primary (CUP) origin. CUPs are characterized by aggressive progression and poor prognosis [10], accounting for 3–5% of all cancers worldwide [11]. Pathology assessment, which plays a key role in determining primary cancer types, is often lacking for these tumors and can be a challenging task for highly metastatic or poorly differentiated tumors. Currently, established targeted therapies are lacking for CUPs. Therefore, there is a need to develop a tool that is able to guide the classification of CUPs and help deliver more accurate primary cancer-type predictions for patients. In order to classify CUPs, Moon et al. recently developed the Oncology NGS-based Primary Cancer-Type Classifier (OncoNPC, 2023), an XGBoost-based classifier [12]. Their key hypothesis is that the genomic signatures, age, and sex of patients encode the information needed for the accurate classification of CUPs. They trained OncoNPC on targeted next-generation sequencing (NGS) from 36,445 tumors across 22 cancers to identify specific cancer types and unknown primary tumor cancer types (i.e., CUPs). Using a variety of genomic measures (mutations, mutational signatures, copy number alterations) and metadata, including patient age at the time of sequencing and sex, allowed for accurate cancer type prediction. The authors also hypothesized that the classified CUP cancer types would exhibit increased polygenic germline risk for the corresponding cancers, and that these predictions could be used for further risk stratification and to potentially guide treatment decisions. Importantly, this study also showed that patients who prospectively received treatments concordant with their OncoNPC-predicted cancer types exhibited significantly better survival outcomes than those who received discordant treatments.

While this study demonstrated the benefits of the accurate classification of unknown tumors through OncoNPC, there are some future directions. Although the current study primarily used retrospective electronic health record (EHR) data from a single institution for downstream clinical analyses, OncoNPC may offer good generalizability to a broader patient population. Future studies could benefit from larger, more diverse datasets. Incorporating more diverse patient populations could help enhance the model’s performance across different races and ethnicities.

### 2.2. Patient Stratification

The accurate grouping of cancer patients into respective cancer types (or subtypes) and staging is crucial for devising better targeted therapies and management in the clinic. However, patient stratification is a not an easy task as there are few reliable genetic or molecular markers that can accurately stratify patients due to their diverse heterogeneity. Hence, signals from higher biological hierarchies such as pathways and molecular processes might provide better information to robustly stratify cancer patients.

Elmarakeby et al. recently developed a biologically informed deep neural network, called P-Net (2021), to stratify prostate cancer patients [13]. The design of the P-Net algorithm is built upon the hypothesis that evolutionarily conserved biological interactions and hierarchical structures and information flows can be recapitulated by deep neural network architectures. In P-Net, the multi-omic status of genes (mutations, copy number, methylation, and expression) is used as input. The connection of input nodes is determined by the involvement of a gene (input node) in each pathway (a node in the first hidden layer). If the pathway contributes to a biological process that is relevant to the disease, the connection continues to the next hidden layer node (the biological process) and then finally reaches the output node, representing a disease state. Once the model is trained, researchers can trace back hidden nodes corresponding to pathways or biological processes and thus identify key pathway-representing nodes whose activation significantly contributes to disease prediction. In this way, P-Net additionally offers an interpretable computational platform for researchers to identify the critical biological pathways and associated genes expression profiles contributing to disease classification.

Although curated pathway databases are incredibly useful, our knowledge on the specific gene–pathway relationship is incomplete; thus, the neural network architecture of P-Net might leave out a number of important genes or pathways that contribute to disease stages. However, the work by Elmarakeby et al. demonstrates the flexibility of deep neural networks in their architectural design, particularly the inclusion of specific domain knowledge, a strategy that shows great potential in other areas of cancer research such as finding driver mutations in cancer development.

### 2.3. Deciphering New Class of Cancer Genes

Cancer-driving genes are not limited to mutated oncogenes, but can involve a myriad of genes whose function does not directly involve tumorigenesis but that are needed for tumor maintenance. This includes the so-called “never mutated” oncogenes. A previous study by Gatenby et al. used a computational modeling approach to reveal the clinical benefits of targeting these never mutated oncogenes [14]. Building upon these observations, we proposed that there is a new class of cancer genes, referred to as “dark cancer genes” or Class II cancer genes, that are neither mutated nor differentially expressed but act as “signal linkers” to coordinate oncogenic signals between mutated and differentially expressed genes. These cancer-relevant genes are often missed using traditional statistical methods.

To detect dark cancer genes, we developed the Machine Learning-Assisted Network Inference (MALANI) tool (2017) [15]. The MALANI algorithm was designed based on the hypothesis that the dot product of gene expression, regardless of its mutational status, captures nonlinear information that reflects the importance of a given gene pair in cancer etiology. The MALANI algorithm was implemented using support vector machines (SVM) and trained across nine cancer types. Supervised learning (cancer versus normal) was performed for each cancer type, using the dot product of gene–gene expression pairs as inputs. Feature selection methods were employed to identify gene pairs whose dot products improved the classification performance. Cancer-specific networks were reverse engineered from selected gene pairs and these successfully identified dark cancer gene candidates that are not differentially expressed or mutated (approximately 3% of the ~19,000 genes). Dark cancer genes were also found to function as coordinators between differentially expressed network hub genes, which have high connectivity in a network, and highly mutated genes. The MALANI algorithm therefore can identify linker genes that are important in conveying oncogenic signals and offers a strategy for new targeting opportunities.

However, there is still room to improve MALANI. For instance, the MALANI algorithm can be devised to incorporate other omics layers such as epigenetics and proteomics. Also, it remains to be seen how cancer progression affects the type and number of dark cancer genes. Future work is needed to explore how targeting dark cancer genes can better benefit cancer patients.

### 2.4. Finding Chemical Fingerprints That Associate with Drug Response

Predicting how cancer cells respond to a given drug is another important area in oncology. In particular, drug prediction and drug matching play a pivotal role in personalized medicine and are key to devising effective therapeutic interventions. Due to the intricate nature of pharmacokinetics (how bodies modulate drug actions) and pharmacodynamics (how drugs interact with bodies), specific treatments for individual patients are becoming more and more important for patients’ pharmacological outcome. Identifying the right combination of biological features underlying personal cancer etiology is crucial for predicting how a patient will respond to specific drugs. Precision in drug prediction thus not only enhances treatment efficacy, but can also minimize potential side effects.

Symbolic regression (SR), which is commonly built in combination with a genetic algorithm, is a regression method guided by the hypothesis that the right combinations of symbolic features (e.g., mathematical operators) encode crucial information to govern the behavior of a system, such as drug response and cancer etiology. Symbolic regression has been successfully used to discover physical laws that govern the properties of physical systems [16,17], by searching the equation space to find the best mathematical function that fits the data. Some examples are the TuringBot software (v2.16.1) and AI Feyman algorithm [18]. In summary, this approach offers a promising avenue for distilling complex biological relationships into interpretable simple rules, bridging the gap between intricate molecular interactions and actionable insights for therapeutic interventions and drug response.

A recent study described a method for generating interpretable quantitative structure–activity relationship models in the field of chemoinformatics. The tool used in this research is called “Filter-Introduced Genetic Programming” (FIGP) (2022) [19]. FIGP is an extension of symbolic regression (SR) combined with Genetic Programming (GP). The goal is to discover mathematical expressions that can describe the relationships within a dataset. The underlying hypothesis of this method is that by incorporating three filters into GP-based SR (Function filter, Variable filter, and Domain filter), along with nonlinear least-squares optimization, it is possible to improve the predictive ability of SR models while generating simpler and more interpretable mathematical expressions. The authors propose that FIGP will be particularly useful for generating interpretable quantitative structure–activity relationship/quantitative structure–property relationship (QSAR/QSPR) models. The study provides a detailed explanation of the methodology and the experimental conditions for both FIGP and conventional GP. It also outlines the evaluation metrics used to assess the predictive performance of these models.

### 2.5. Extracting Associations of Data Attributes That Explain Data Properties

The functional associations between genes are important in understanding cancer etiology, in addition to the physical interactions between proteins and other gene products, such as RNAs. Conventionally, functional associations between genes are inferred from statistical-based correlative approaches, such as the co-expression of genes using Pearson’s correlation [20] and mutual information [21]. However, conventional statistical methods often fail to capture the nonlinear functional associations of genes that explain the properties of high-dimensional cancer sequencing data.

We recently developed the Artificial Neural Network Encoder (ANNE) (2022), an Artificial Neural Network (ANN) implemented with a novel weight engineering algorithm to reverse engineer gene–gene interactions from gene expression data [22]. The weight engineering algorithm was inspired by how the human brain learns and stores knowledge in the form of sparse spatial representations. One way the brain encodes information is by leveraging connections between neurons, which are fundamentally plastic and select which information to store based on its importance. This process implies that changes such as “pruning” occur during the learning process, i.e., redundant information is discarded. This process allows the learned information to be sparsely represented and distributed as “weights” in inter-neuronal connections throughout the neocortex. The underlying hypothesis of this weight engineering algorithm is that inter-neuronal weights resulting from the learning process represent the knowledge the brain learns from observation and data processing. We hypothesized that a similar scenario also takes place in ANN. Hence, mathematical manipulation of trained inter-neuronal weights might be able to recover the learned knowledge by ANN. We tested this idea using an autoencoder, a commonly used deep learning model able to reduce data dimensionality via the reconstruction of the input data [23]. Here, the autoencoder represents a “little brain” and the trained inter-neuronal weights represent the “knowledge” learned. Using a breast cancer cohort as a proof-of-concept study case, we showed that by reverse engineering gene–gene association networks (the extracted “knowledge” from trained autoencoders) with our weight engineering algorithm, we were able to identify both known and novel clinical aspects of breast cancer etiology.

Our weight engineering algorithm shows that trained ANNs indeed encode learned “knowledge”, which is represented by associations between data attributes (e.g., gene expression) in inter-neuronal weights. These associations between data attributes can serve as a knowledge discovery framework to uncover the novel functional associations between genes that underpin cancer etiology. Yet, there is still plenty of room to improve knowledge discovery via weight engineering, for instance, by incorporating multi-omics data to decipher how genes are regulated at multi-omics levels.

### 2.6. Phenotype Prediction

Even patients diagnosed with the same cancer type can exhibit distinct cancer phenotypic properties, such as tumor aggressiveness and responsiveness to therapeutics. Cellular phenotypes are not the outcomes of single gene activity, but involve a myriad of genes and pathways at different functional hierarchies within the cells. Although various AI algorithms including deep learning methods have been employed to perform phenotype prediction, most AI algorithms, especially ANNs, are “black boxes” and determining how AI models are trained to make decisions is difficult [24].

Ma et al. devised an ingenious deep learning algorithm called DCell (2018) to visualize the “black box” of deep learning [25]. The underlying hypothesis of DCell is that, by incorporating an extensive collection of domain knowledge and data about cellular subsystems and their hierarchical structure, it is possible to create an interpretable deep neural network that can accurately simulate the function of a eukaryotic cell. This methodology overcomes a current limitation of ANNs by incorporating extensive knowledge of cell biology, making it a visible neural network (VNN) with a more interpretable inner structure. Through a study on the budding yeast *Saccharomyces cerevisiae* with a focus on accurately simulating cellular growth, a complex phenotype influenced by genetic interactions, Ma et al. showed that DCell could successfully capture the phenotypic variation of cellular growth in yeast, including the non-additive portions arising from genetic interactions.

A more recent study by Kuenzi et al. using DCell (2020) aimed to predict drug response phenotypes and drug synergy in human cancer cell lines revealed the power of this algorithm in constructing interpretable models for cancer treatment [26]. In this study, Dcell was integrated with three embeddings as inputs: drug response, genotype, and drug structure. The authors envision that future work may integrate mutations with additional levels of molecular information such as epigenetic states, gene expression, or microenvironmental influences. Since current Dcell structures are built upon both annotated gene ontologies (GO) and curated literature that can be biased to well-studied genes, the algorithm can be enhanced by integrating resources from novel gene–gene associations or gene–function assignments from computational models with robust modeling results.

### 2.7. Uncover New Class of Genes That Govern Spatial Organization of Cells in Tumor Microenvironment

In addition, the gene activities taking place within cancer cells that sustain their functional and survival fitness, their extracellular environments or tumor microenvironments (TMEs) are equally important in shaping their phenotypic behaviors [27]. As a consequence, the spatial organization of cells within a TME can be influenced by its neighboring cells [28]. With advances in spatially resolved sequencing technologies in recent years, researchers can now map cells at their spatial locations in unprecedented detail [29]. A number of statistical-based and AI methods have been developed to identify spatially variably expressed genes (SVGs) [30,31,32] and understand spatial cell distribution. Though it is important to indicate differential gene activities across distinct TME regions, SVGs cannot indicate how these gene activities affect the spatial organization of cells. This is because the detection of SVGs is, in principle, similar to the identification of differentially expressed genes detected from bulk or single cell sequencing data; though it provides valuable information, it cannot explain the underlying regulatory processes that dictate the organization of cells in their microenvironments.

The success of the de novo in silico reconstruction of organization of cells from transcriptomics data [33,34] implies that whole-genome expression encodes gene activities that indicate how cells are arranged in space. As such, we hypothesize the existence of a new class of genes named spatially predictive genes (SPGs), whose collective expression can predict where cells (or subpopulations of cells) are organized in space. This inspired us to develop a novel deep learning algorithm called Spatially Informed AI (SPIN-AI) (2023) to test this hypothesis [35]. SPIN-AI employs an unbiased approach and uses only the spatial gene expression, per patient per slide, as an input and is trained to predict the x and y spatial coordinates in a spatial transcriptomic slide. Using human squamous cell carcinoma [36] as a proof-of-concept study, we showed that the identities of SPGs can be distinct from SVGs, and that their activities can help dictate how cells are spatially arranged in cellular niches. Further, this study proposes that SPGs can be viewed as new actionable targets in cancer treatment. However, there are still a number of areas to be explored. For example, how does the genetic heterogeneity of cancer cells affect the number and identities of SPGs. Another area to investigate is whether SPGs can also inform drug response phenotypes.

## 3. Discussion

With examples of hypothesis-driven AI predecessor tools and their application in distinct areas of oncology, it becomes evident that this new class of AI methods holds great promise for the future of medicine, as shown in Appendix A. However, not all examples discussed above are hypothesis-driven AI in the strict sense. For example, OncoNPC and MALANI do not possess all the desired aspects given in Table 1, but nonetheless offer a transition from conventional to hypothesis-driven AI.

Despite their wide area of application in cancer research, hypothesis-driven AI tools show several areas for improvement. While some models such as OncoNPC have shown efficacy, there is a need to generalize these models across different patient populations and cancer types. However, since these models were developed based on disease-specific domain knowledge, this might inadvertently limit their utility in other disease contexts. Utilizing domain knowledge agnostic to a specific disease type while retaining the flexibility to adapt to a specific biological context can enhance the generalizability of a hypothesis-driven AI tool to broader disease types. Moreover, as the structure and dimensionality of data become more complex by including distinct biological layers such as genetics and environmental, researchers face greater challenges in formulating hypotheses that can effectively manage different levels of data attributes, and this demands further abstract reasoning from researchers. Additionally, the selection of which knowledge domain to integrate into algorithmic design remains subjective and varies among researchers.

Figure 4 summarizes the new ingredients needed for designing hypothesis-driven AI algorithms and illustrates how these can be utilized to unearth hidden gems embedded within the data to help better inform clinical decisions. Below we provide some open questions and food for thought that we think could inspire the development of hypothesis-driven AI in the near future.

## 4. Food for Thought

Given the areas of development, a strategic way to dissect different layers of data attributes and extract information relevant to the hypothesis is vital for the evolution of hypothesis-driven AI tools. In the following sections, we propose several directions for future development in this context.

All AI methods, including hypothesis-driven AI, involve learning from data and in that sense are data driven. A key distinction of hypothesis-driven AI lies in the fact that it opens a new way of thinking regarding the extraction of the hidden information embedded in the data, which cannot be readily accomplished by conventional AI. This framework shapes how data will be collected and even generated given the scientific or medical hypothesis to be tested, for example, understanding how cancer develops. Another example is how to transform data attributes (e.g., genetic mutations) into input forms that are compatible with the proposed hypothetical frameworks. This, in fact, will likely allow researchers to use data in more efficient and creative ways.

One notable aspect is the significance of altered epigenetic landscapes in cancer etiology, which goes beyond genetic mutations [37,38]. In addition to DNA methylation and histone modification, it is now becoming appreciated that alterations in epigenetics can lead to the gain of super-enhancers [39] and gross changes in chromatin remodeling [40]. Further, the processes of epigenetic regulation can go beyond the cellular level, thereby impacting the tissue, organ, and even the whole-body level. We recently proposed the Manifold Epigenetic Model (MEMo), a conceptual framework able to explain how epigenetic memories can be established at the body-wide level [41]. We defined manifold epigenetics as a study that is concerned with the totality of molecular, cellular, and environmental systems-based mechanisms that confer body-wide phenotypic memories without altering DNA sequences. Guided by this concept, it is possible to devise hypothesis-driven AI algorithms that are analogous to P-Net, DCell, and ANNE by taking multi-layered epigenetic regulation into consideration.

Another biological concept we proposed recently is the Gene Utility Model (GUM), which states that the significance of a gene in disease etiology hinges on its utility within a protein–protein interaction (PPI) network specific to a certain disease context [42]. Here, we simulated gene utilities using a process-guided algorithm [43] and showed that gene utility profiles capture the patterns of chromosomal aberrancies in advanced-stage neuroblastoma, a childhood cancer with few somatic mutations but that shows enigmatic conserved chromosomal abnormalities [44]. Hence, the integration of biological concepts, such as gene utility, with omics profiles in the formulation of new-generation hypothesis-driven AI algorithms can help AI algorithms to sift through complex omics and guide the identification of targetable associations.

Cancer can also be described as a type of chronic disease that is similar to metabolic syndromes and neurodegenerative diseases [45] in the sense that many patients require long-term and complex care. As chronic diseases often involve multiple organs and have high comorbidity rates with other diseases, it is important to broaden our perspective on cancer etiology by examining the modulation of multi-organ functions. For instance, kidney injury has been linked to cachexia [46], and dysfunctional interorgan crosstalk can also induce pathological systemic niches and contribute to disease progression, including cancer metastasis [47]. We recently proposed the Locked-State Model (LoSM), which states that positive feedback loops sustaining inter-organ communication can help build memory-like properties that “lock” healthy and disease states [48]. LoSM provides a new conceptual framework to describe how memory-like inter-organ communication can contribute to disease etiology and how therapeutic intervention on pathological organ crosstalk can provide pharmacological benefits to patients. The concept of LoSM therefore encourages the design of AI algorithms based on hypotheses that incorporate not only cell–cell communication, but also organ–organ communication.

Drugs are often not ideal “magic bullets” that only target a single molecule [49]. Rather, drugs are promiscuous and act on multiple targets, albeit with different binding specificities, and often lead to off-target effects [50]. Hence, the action of drugs is often multifaceted and multidimensional, involving gene products that participate in diverse biological pathways and even different organs. For instance, altering the microbiome can affect sensitivity to immune checkpoint inhibitors [51]. It is therefore important to take these factors into account when devising hypothesis-driven AI for the drug discovery pipeline. We previously proposed Manifold Medicine built upon five body-wide vectorial axes (genetic, molecular network, internal environment, neural–immune–endocrine, microbiota) and outlined the manifoldness nature of the mode-of-action of drugs into target modes (subject), regimen modes (predicate), and patient modes (modifier), and illustrate how a manifold treatment, combining drugs with different modes of action, can counteract the vectorial tendencies of diseases [41]. We believe that this conceptual framework could be incorporated in formulating new hypotheses in AI algorithm design to uncover how different types of body axes and the mode-of-action of a drug interact.

Generative AI (GAI) also presents a cutting-edge opportunity to revolutionize cancer research and can be included in the design of hypothesis-driven AI models. For instance, Generative Adversarial Networks (GANs), a generative model, can be employed to simulate realistic biological data including genomic and imaging data, which are particularly valuable when access to large, diverse datasets is limited. In the context of cancer research, generative AI can be used to generate synthetic patient cohorts [52,53]. By combining these synthetic cohorts with real-world data, the robustness and generalizability of current predictive AI models can be increased. For example, GANs can be trained on existing genomic and imaging data to generate synthetic samples that mimic the characteristics of real cancer data. Additionally, generative AI can be utilized to simulate the evolution of cancer over time [54], offering insights into dynamic changes in tumor biology and aiding in the development of more adaptive and personalized treatment strategies. In earlier work, we proposed that gene–gene pairs that flip in their activities can modulate drug response phenotypes [55]. We named these gene pairs “regulostats”. We believe that GAI has the power to encapsulate such relationships and advance our understanding of how phenotypic behaviors are modulated in cancer cells. Thus, it is possible to design generative AI architectures by incorporating domain-specific knowledge and simultaneously incorporate broad biological concepts.

A future AI tool that can use not only multi-omics, but also medical imaging data, liquid biopsy omics, physiological data, and patient self-reported outcomes to track patient outcomes and thus inform on patients’ responses to future treatment could be designed as a software. With more comprehensive data being integrated, AI can be more accurate in predicting the next step in tumor progression across different patients. With respect to the therapeutic targeting, the development of an AI platform that integrates knowledge from diverse sources, including the scientific literature, clinical trial data, and molecular databases, to identify novel drug repurposing opportunities for specific cancer types is also needed. By constructing a dynamic knowledge graph to represent the intricate relationships among drugs, diseases, molecular targets, and biological pathways, this new AI model can utilize pattern recognition to uncover the hidden associations between drugs and signaling pathways. The platform should also incorporate real-world patient data from electronic health records for validation and personalized patient stratification. Through this platform, tumor targeting might become more efficient and allow for advancements in drug repurposing.

## 5. Conclusions

In conclusion, we have defined a new class of AI algorithm called hypothesis-driven AI, which shows great promise in overcoming various challenges in current oncology research. The examples provided in this review make it clear that this new class of AI methods is beginning to flourish. Unlike conventional AI, designing learning algorithms of hypothesis-driven AI requires researchers to demonstrate ingenuity, creativity, innovation, and the careful selection of domain knowledge. Such special features can have dual consequences, representing both the strengths and weaknesses of hypothesis-driven AI methods. The strength is that researchers can formulate hypothetical modes of the gene–gene or gene–pathways associations that underpin cancer etiology and develop learning algorithms to perform “AI thought experiments” to validate the proposed hypotheses. This will enable the discovery of novel gene–gene and gene–pathway associations, which are often overlooked by conventional AI. On the other hand, it is challenging for researchers to hypothesize all generalizable modes of relationships.

Nevertheless, by aligning computational methodologies with well-informed hypotheses, hypothesis-driven AI offers a targeted and informed way to address issues ranging from tumor detection to drug targeting. Since the design of these AI algorithms is driven by hypotheses, researchers can leverage the synergy between domain knowledge, computational models, and experimental validation to gain a deeper understanding of cancer biology and develop more effective cancer treatments in the near future. Through the exploration of key concepts, methodologies, and promising applications, we believe this review will serve as a roadmap towards the development of the next generation of hypothesis-driven AI to advance individualized and precision medicine.

## Figures and Tables

**Figure 1 cancers-16-00822-f001:**
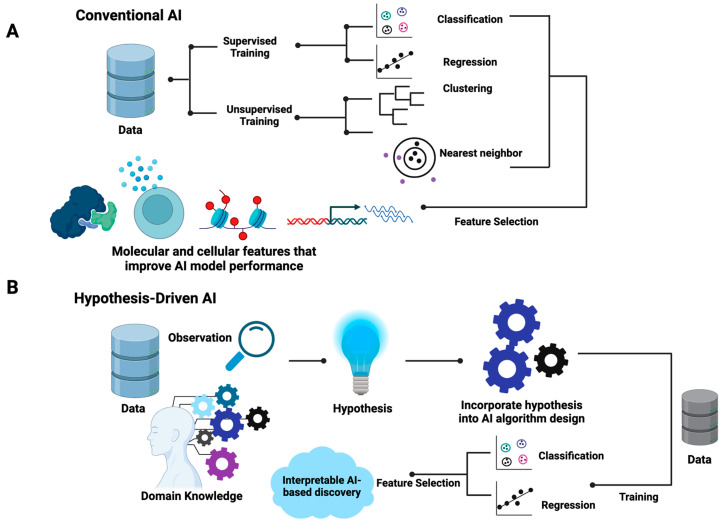
Comparison of conventional AI and hypothesis-driven AI. (**A**) The learning pipeline of conventional AI. The learning algorithms often include weighted connections and distance metrics without the need to include existing domain knowledge or a hypothesis into the design of learning algorithms. (**B**) The design and learning pipeline of hypothesis-driven AI. Knowledge or hypothesis are the built-in components in the design of learning algorithms; these facilitate the discovery of novel associations between attributes that can explain data properties.

**Figure 2 cancers-16-00822-f002:**
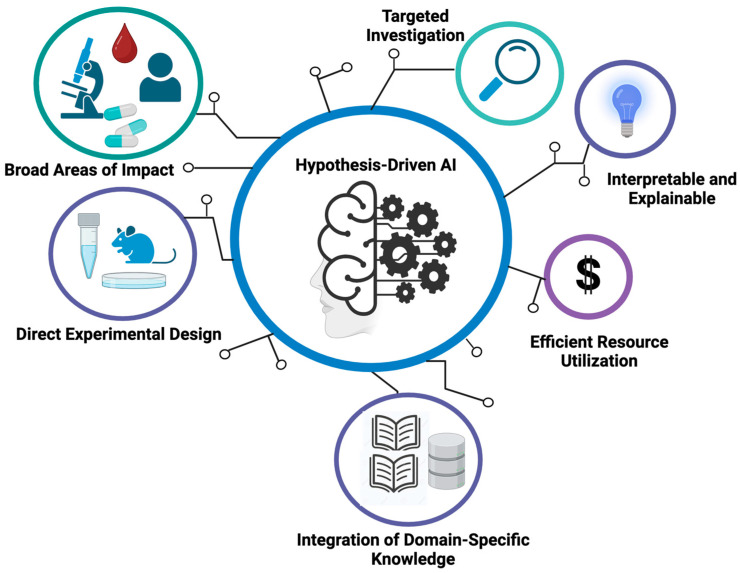
Key characteristics of hypothesis-driven AI methodologies.

**Figure 3 cancers-16-00822-f003:**
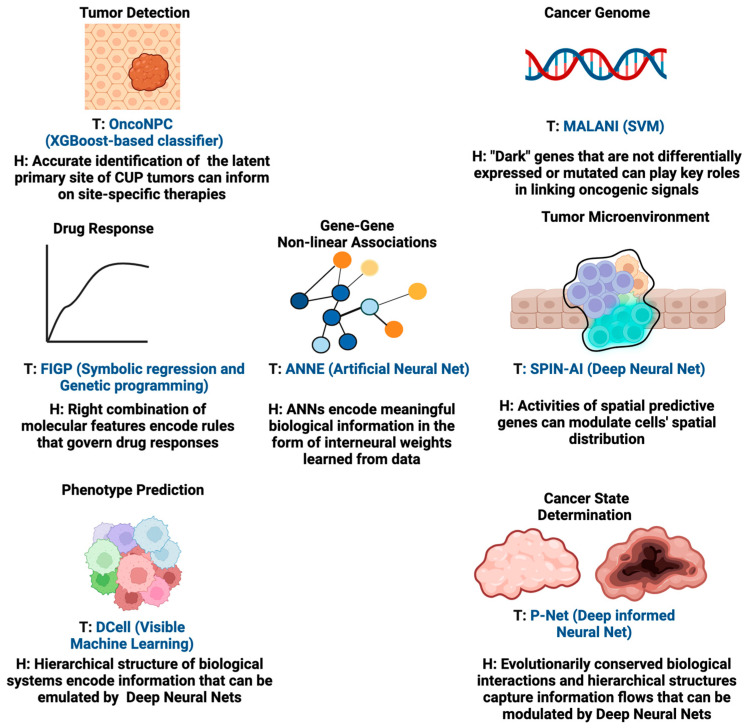
Examples of tools that are similar to hypothesis-driven AI in different areas of oncology research. The design of the learning algorithms to a certain extent aligns with the configuration of hypothesis-driven AI. The underlying hypothesis (H) of each algorithm tool (T) is shown in the figure.

**Figure 4 cancers-16-00822-f004:**
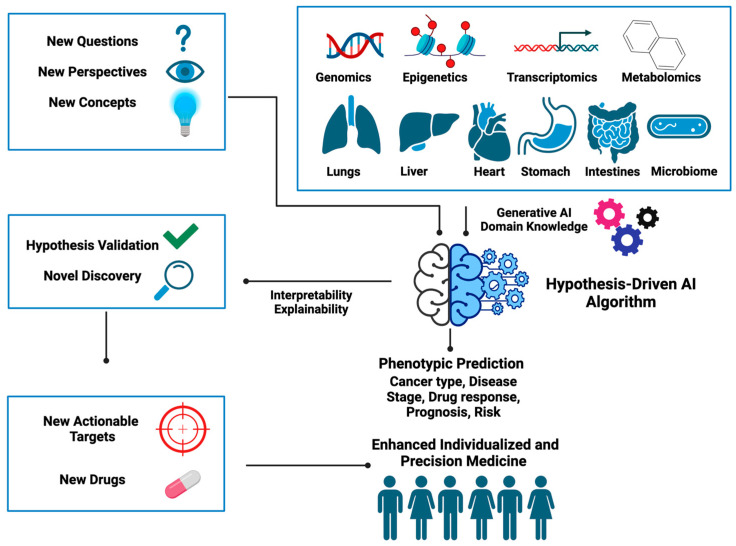
Key ingredients and areas of application in the design of next-generation hypothesis-driven AI.

**Table 1 cancers-16-00822-t001:** Comparison between conventional AI and hypothesis-driven AI methods.

Characteristics	Conventional AI	Hypothesis-Driven AI
Focus	Broad exploration of data patterns	Targeted investigation of specific hypotheses
Interpretability	Less interpretable	More interpretable
Resource Efficiency	May require extensive data and computational resources	Uses resources more efficiently by focusing on specific areas of interest
Integration of Domain-Specific Knowledge	Limited	Encourages integration of domain-specific knowledge for meaningful biological insights
Experimental Validation	May not explicitly incorporate mechanistic insights for experimental validation follow-up	Test hypotheses and incorporates valid experimental designs to confirm casual results or associations

## Data Availability

All data are publicly available.

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
