# Peer review of "The Rise of Hypothesis-Driven Artificial Intelligence in Oncology"

_cancers, 2024, doi:10.3390/cancers16040822_

Round 1

Reviewer 1 Report

Comments and Suggestions for Authors

1.This review introduces a new class of AI algorithms —hypothesis-driven AI,which holds great promise to discover new mechanistic frameworks that explain the complexity of cancer etiology and improve treatment regimens for individual patients,has great application prospects.

2. Suggest merging "1.2. Challenges in Oncology Research" into "1. Introduction" and introduce the advantages and disadvantages of conventional AI in Oncology.

3.Table 1 adopts a three line table format.

Comments on the Quality of English Language

Minor editing of English language required.

Author Response

Manuscript ID: cancers-2796700

Type of manuscript: Review

Title: The Rise of Hypothesis-Driven Artificial Intelligence in Oncology by Xiangu et al.

Overall Response: We are grateful to both reviewers for their enthusiasm of our review manuscript and their overall constructive suggestions. We have carefully revised our manuscript as suggested which greatly improved the overall content. Additionally, the text was thoroughly revised by our native English-speaking colleagues and approved by a professional English editor.

Reviewer #1

1.This review introduces a new class of Al algorithms -hypothesis-driven Al, which holds great

promise to discover new mechanistic frameworks that explain the complexity of cancer etiology and improve treatment regimens for individual patients, has great application prospects.

Response: We thank the reviewer for his/her enthusiasm for our proposed new class of AI algorithms and its great potential in advancing cancer research.

  1. Suggest merging "1.2. Challenges in Oncology Research" into "1. Introduction" and introduce the advantages and disadvantages of conventional Al in Oncology.

Response: We thank the reviewer for this constructive suggestion. In the revised manuscript we have merged “Challenges in Oncology Research” with the Introduction and now include advantages and disadvantages of conventional Al in oncology. This indeed improved the overall flow of the Introduction.

3.Table 1 adopts a three-line table format.

Response: We thank the reviewer for this kind suggestion. We have updated the table formatting as suggested.

Minor editing of English language required.

Response: We have revised the text thoroughly and corrected grammar and typos with the help of a professional English editor. Our revised manuscript has also been read and approved by our native English-speaking colleagues.

Reviewer 2 Report

Comments and Suggestions for Authors

Major Comments:

1. Are there controversies in this field? What are the most recent and important achievements in the field? In my opinion, answers to these questions should be emphasized. Perhaps, in some cases, novelty of the recent achievements should be highlighted by indicating the year of publication in the text of the manuscript.

2. The discussion section is modest.

3. Abstract: not properly written.

4. Conclusion: The section devoted to the explanation of the results suffers from the same problems revealed so far. Your storyline in the results section (and conclusion) is hard to follow. Moreover, the conclusions reached are really far from what one can infer from the empirical results.

5. The discussion should be rather organized around arguments avoiding simply describing details without providing much meaning.

6. Figures not clear enough.

7. Limitation of The Rise of Hypothesis Driven Artificial Intelligence in Oncology must need to write in details.

7. Spacing, punctuation marks, grammar, and spelling errors should be reviewed thoroughly. I found so many typos throughout the manuscript.

8. English is modest. Therefore, the authors need to improve their writing style. In addition, the whole manuscript needs to be checked by native English speakers.

Comments on the Quality of English Language

English is modest. Therefore, the authors need to improve their writing style. In addition, the whole manuscript needs to be checked by native English speakers.

Author Response

Manuscript ID: cancers-2796700

Type of manuscript: Review

Title: The Rise of Hypothesis-Driven Artificial Intelligence in Oncology by Xiangu et al.

Overall Response: We are grateful to both reviewers for their enthusiasm of our review manuscript and their overall constructive suggestions. We have carefully revised our manuscript as suggested which greatly improved the overall content. Additionally, the text was thoroughly revised by our native English-speaking colleagues and approved by a professional English editor.

Reviewer #2

  1. Are there controversies in this field? What are the most recent and important achievements in

the field? In my opinion, answers to these questions should be emphasized. Perhaps, in some cases,

novelty of the recent achievements should be highlighted by indicating the year of publication in

the text of the manuscript.

Response: We thank the reviewer for these very thoughtful comments. In the second paragraph of Discussion section (page 11) we now added a section on current controversies in this AI field. We have also included the year of publication for hypothesis-driven AI methods whenever these are first introduced in the text.

  1. The discussion section is modest.

Response: We have expanded our Discussion section by including a paragraph that introduces limitations and key ingredients needed to enhance the next-generation of hypothesis-driven AI in the near future.

  1. Abstract: not properly written.

Response: We have rewritten and streamlined the “Abstract” following journal guidelines while providing a broad overview of the manuscript.

  1. Conclusion: The section devoted to the explanation of the results suffers from the same problems

revealed so far. Your storyline in the results section (and conclusion) is hard to follow.

Moreover, the conclusions reached are really far from what one can infer from the empirical

results.

Response: We have expanded our “Conclusion” section, emphasizing challenges needed to overcome while highlighting the great promise of hypothesis-driven AI over conventional AI in guiding novel cancer discovery.

  1. The discussion should be rather organized around arguments avoiding simply describing details

without providing much meaning.

Response: We thank the reviewer for this constructive suggestion. We have carefully reorganized the discussion and included additional clarifications wherever applicable. We also consulted our colleagues without AI background for clarity on writing and manuscript overall flow.

  1. Figures not clear enough.

Response: We have increased the font size and resolution of all figures and added minor alterations as needed.

  1. Limitation of The Rise of Hypothesis Driven Artificial Intelligence in Oncology must need to

write in details.

Response: We now have included a more detailed discussion for hypothesis-driven AI and its development and limitations in the revised “Discussion” section (page 11).

  1. Spacing, punctuation marks, grammar, and spelling errors should be reviewed thoroughly. I found so many typos throughout the manuscript.

Response: We thank the reviewer for this very kind suggestion. We have rewritten sentences and corrected grammar and typos. Each author has carefully revised the text to ensure all misspellings were corrected.

  1. English is modest. Therefore, the authors need to improve their writing style. In addition, the

whole manuscript needs to be checked by native English speakers.

Response: We have corrected any detectable grammar mistakes with the help of a professional English editor. Our manuscript has also been read and approved by our native English-speaking colleagues.

Round 2

Reviewer 2 Report

Comments and Suggestions for Authors

1. Data-Driven vs. Hypothesis-Driven AI: This section needs to add.

2. Examples of how AI is used to test specific hypotheses and generate novel insights.

3. Discussion on how hypothesis-driven AI interfaces with experimental research in oncology; need to include.

4. Clinical Applications are also missing.

Comments on the Quality of English Language

Moderate editing of English language required

Author Response

Manuscript ID: cancers-2796700

Type of manuscript: Review

Title: The Rise of Hypothesis-Driven Artificial Intelligence in Oncology by Xianyu et al.

Reviewer #2 – Re-re-submission

Overall Response: We sincerely thank Reviewer #2 for his/her feedback and suggestions to further strengthen our manuscript. We have carefully revised our manuscript and the revised text was thoroughly reviewed by our native English-speaking colleagues and approved by a professional English editor.

  1. Moderate editing of English language required.

Response: We have carefully edited our text to avoid any grammatical errors and re-checked sentences that could benefit from additional clarification. Our manuscript has been read and approved by our native English-speaking colleagues and a professional English editor.

  1. Data-Driven vs Hypothesis-Driven AI: this section needs to add.

Response: We thank the reviewer for this suggestion. We have revised the section “What is Hypothesis-Driven AI” to clearly identify the differences between Data-Driven AI and Hypothesis-Driven AI (lines 103-131). Additionally, our Table 1 and Figure 1 compare and contrast these very distinct AI approaches. Finally, we have extended this discussion under the section “Food for thought” (lines 503-512).

  1. Examples on how AI is used to test specific hypotheses and generate novel insights.

Response: We thank the reviewer for this great suggestion. We have included examples of how AI can be used to test specific hypotheses and generate novel insights by citing different AI tools, including their underlying hypotheses, and their findings in subsections of “Recent Developments and Application of Hypothesis-Driven AI in Oncology” across different oncology areas. We also provide additional insights in the section “Food for thought”.

  1. Discussion on how Hypothesis-Driven AI interfaces with experimental research in oncology; need to include.

Response: We thank the reviewer for this important insight. We provide several examples of Hypothesis-Driven AI alike methodologies in the section “Why Do We Need Hypothesis-Driven AI in Oncology Research” and emphasize their interface with experimental research. We have also included this point in the revised section of “Food for thought” (lines 503-512).

  1. Clinical applications are also missing.

Response: We thank the reviewer for this comment. We have highlighted the clinical applications of Hypothesis-Driven AI by describing recently published state-of-the-art medical hypotheses in the section “Food for thought”. For example, in this section we also highlight the use of Hypothesis-Driven AI to decipher: 1) the role of organ-organ communication and multi-organ function in cancer etiology; 2) identify mechanisms that drive organ and whole-body memory and 3) manifold nature of drugs and their impact on devising better targeting strategies.

Round 3

Reviewer 2 Report

Comments and Suggestions for Authors

Authors addressed all of my comments. The revised manuscript can be accepted for final publication.

Comments on the Quality of English Language

Minor revisions